# Improved Group Robustness via Classifier Re-training on Independent Splits

## Abstract

Deep neural networks learned by minimizing the average risk can achieve strong average performance, but their performance for a subgroup may degrade, if the subgroup is underrepresented in the overall data population. Group distributionally robust optimization (Sagawa et al., 2020a, GDRO) is a standard baseline for learning models with strong worst-group performance. However, GDRO requires group labels for every example during training and can be prone to overfitting, often requiring careful model capacity control via regularization or early stopping. When only a limited amount of group labels is available, Just Train Twice (Liu et al., 2021, JTT) is a popular approach which infers a pseudo-group-label for every unlabeled example. The process of inferring pseudo labels can be highly sensitive during model selection. To alleviate overfitting for GDRO and the pseudo labeling process for JTT, we propose a new method via classifier retraining on independent splits (of the training data). We find that using a novel sample splitting procedure achieves robust worst-group performance in the fine-tuning step. When evaluated on benchmark image and text classification tasks, our approach consistently reduces the requirement of group labels and hyperparameter search during training. Experimental results confirm that our approach performs favorably compared with existing methods (including GDRO and JTT) when either group labels are available during training or are only available during validation.

## 1 Introduction

For many tasks, deep neural networks (DNNs) are often developed where the test distribution is identical to and independent of the train distribution, which can be referred to as IID generalization. The performance of a DNN is also known to worsen when the testing distribution differs from the training distribution. This problem is often referred to as out-of-distribution (OOD) generalization. OOD generalization is crucial in safety-critical applications such as self-driving cars (Filos et al., 2020) or medical imaging (Oakden-Rayner et al., 2020). Hence, addressing the problem of OOD generalization is foundational for real-world deployment of deep learning models.

A notable setting where OOD generalization problems appear is the *group-shift* setting, where different groups of the data may have a distribution shift. In this setting, there are predefined attributes that divide the input space into different groups of interest. Here, the goal is to find a model that performs well across several predefined groups (Sagawa et al., 2020a). Similar to other problems in OOD generalization, DNNs learned by empirical risk minimization (ERM) are observed to suffer from poor worst-group performance despite good average-group performance.

The difficulty with learning group robust DNNs can be attributed to the phenomenon of shortcut learning (Geirhos et al., 2020) or spurious correlation (Sagawa et al., 2020a; Arjovsky et al., 2019). Shortcut learning poses that ERM favors those models that discriminate based on simpler and/or spurious features of the data. However, one wishes for the learning algorithm to produce a model that uses features (i.e. correlations) that performs well not only on the train distribution, but also on all potential distributions that a task may generate, like that of a worst-group distribution.

In recent years, the group-shift setting has received considerable attention, where Sagawa et al. (2020a) first investigates distributional robust optimization (DRO) (Duchi et al., 2021; Ben-Tal et al., 2013) in this setting and introduces *Group DRO* (GDRO) that attempts to directly optimize for the worst-group error. Since then, GDRO has been the standard method for producing group-robust

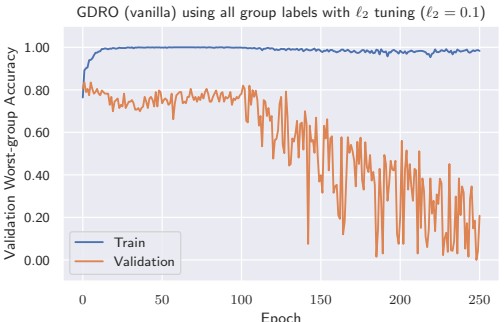 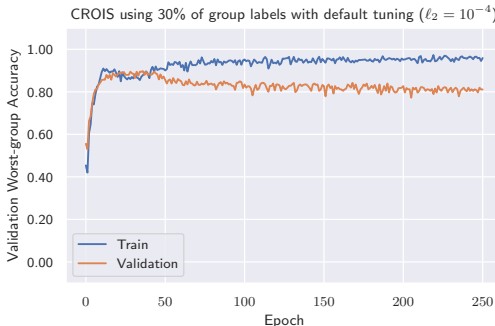

Figure 1: Worst-group learning curve on the Waterbird dataset between GDRO (**left**) and CROIS (**right**) from the setting in Section 4.2. In the left figure, the validation accuracy of GDRO becomes unstable as the number of epochs increase beyond 100 while the training accuracy remains close to 100%. Our approach CROIS instead uses 30% of the training data with group labels for fine-tuning the classification layer of a DNN that is obtained via ERM using the rest of the training data without group labels (see also Algorithm 1). This allows CROIS to reuse ERM features while improving generalization compared with GDRO.

models. However, GDRO can be sensitive towards model capacity (Sagawa et al., 2020a) and require group labels for all examples at training time.

As group annotations can be expensive to obtain, many works consider ways to reduce the amount of groups labels needed (Liu et al., 2021; Creager et al., 2021; Nam et al., 2022). These methods usually follow the framework of first inferring pseudo-group-labels using a certain referenced model (pseudo-labeling) and then applying a group-robust algorithm like GDRO on the pseudo-labelled data. While results for these methods have been promising, they usually introduce several more sensitive hyperparameters that can be expensive to tune. Our work aims to obtain group robust models using as few group labels as possible while alleviating the need to carefully control model capacity.

**Our contribution.** In this work, we propose a simple approach, called CROIS. By foregoing the error-prone and costly pseudo-labeling phase to instead concentrate on efficiently utilizing group labels by applying them *only* to the final classifier layer, CROIS achieves good robust performance *without* relying on a multitude of hyperparameters and large scale tuning, which has been a growing concern in the community (Gulrajani and Lopez-Paz, 2021).

In short, CROIS takes advantage of the features learned by ERM (Kang et al., 2019; Menon et al., 2021) while overcoming the deficiency of its memorization behavior (Sagawa et al., 2020b) by utilizing the training data as two independent splits: one *group-unlabeled* split to train the feature extractor and one *group-labeled* split to retrain only the classifier with a robust algorithm like GDRO. We demonstrate through ablation studies that the use of independent splits is crucial for robust classifier retraining. Furthermore, CROIS's restriction of GDRO to only a low-capacity linear layer reduces GDRO's sensitivity towards model capacity as well as the amount of data needed for GDRO to generalize well (e.g. Figure 1 and Figure 3).

For various settings where group labels are only partially available during training (Section 4.1), our experimental results on standard datasets including Waterbird, CelebA, MultiNLI, and CivilComments show improved performance over existing methods including JTT (Liu et al., 2021) and SSA (Nam et al., 2022), despite minimal parameter tuning and no reliance on pseudo labeling. In another setting where more group labels are available (Section 4.2), even when using only a fraction of the available group labels, our competitive robust performance against GDRO demonstrates our method's label efficiency. Finally, our results provide further evidences of ERM trained DNNs containing good features on both image classification (Menon et al., 2021) and natural language classification tasks.

### 1.1 RELATED WORKS

There are three main settings for the group-shift problem: (1) full availability of group labels, (2) limited availability of group labels, and (3) no availability of group label. Other related areas include domain generalization and long-tailed classification.

**Full training group labels.** Most methods here revolve around up-weighing minority groups, subsampling minority groups (Sagawa et al., 2020b), or performing GDRO (Sagawa et al., 2020a). Follow-up works include integrating data augmentation via generative model (Goel et al., 2020) or selective augmentation (Yao et al., 2022) to a robust training pipeline.

**Limited access to group labels.** In this setting, the approach of inferring more group labels for the group-unlabeled data remains the most popular. These *pseudo-group-labels* are usually generated by training a referenced model that performs the labeling. For example, (Liu et al., 2021) utilizes a low-capacity model that create groups by labeling whether an example is correctly classified by the referenced model or not. Similarly, works like (Creager et al., 2021; Dagaev et al., 2021; Krueger et al., 2021; Nam et al., 2022) and (Nam et al., 2020) are variants of this approach of inferring pseudo-group-labels. These methods then proceed to use a group-robust algorithm like GDRO (Sagawa et al., 2020a) or Invariant Risk Minimization (IRM) (Arjovsky et al., 2019) to perform robust training on a new network with the newly generated pseudo-group-labels.

**No group labels.** This setting removes the ability to validate as well as knowledge of potential groups. This makes the problem more difficult as it is unclear which correlation to look for during training. Some theoretical works in this space include Hashimoto et al. (2018); Lahoti et al. (2020). Sohoni et al. (2020) proposes a popular empirical approach in this setting and has popularized the pseudo-labeling and retraining approach. This setting is related to *domain generalization*. Gulrajani and Lopez-Paz (2021) shows through mass-scale experiments that most OOD generalization methods do not improve over ERM given the same amount of tuning and model selection criterion.

**Long-tailed classification.** The long-tailed problem concerns with certain classes having significantly fewer training examples than others (see for example Haixiang et al. (2017) or Zhang et al. (2021) for a survey). Some techniques from the long-tail literature, like Cao et al. (2019), has been applied to the group-shift setting to account for the groups imbalances, as in Sagawa et al. (2020a). Insights from applications of representation learning in the long-tail problem (Kang et al., 2019) gives valuable evidences for ERM trained DNNs containing good features which is central for our method.

## 2 PRELIMINARIES

For a classification task $\mathcal{T}$ of predicting labels in $\mathcal{Y}$ from inputs in $\mathcal{X}$, we are given training examples $\{(x_i, y_i)\}_{i=1}^n$ that are drawn IID from some train distribution $\mathcal{D}_{\text{train}}$. In the domain generalization setting, we want good performance on some unknown test distribution $\mathcal{D}_{\text{test}}$ that is different but related to $\mathcal{D}_{\text{train}}$ through the task $\mathcal{T}$. More explicitly, we wish to find a classifier $f$ from some hypothesis space $\mathcal{F}$ using $\mathcal{D}_{\text{train}}$ such that the classification error $L(f) = \mathbb{E}_{x,y \sim \mathcal{D}_{\text{test}}}[f(x) \neq y]$ of $f$ w.r.t. $\mathcal{D}_{\text{test}}$ is low. This framework encapsulates many problems like adversarial robustness, domain adaptation, long-tail learning, few-shot learning, and the problem considered here: group-shift.

In the group-shift setting (Sagawa et al., 2020a), we further assume that associated with each data point $x$ is an *attribute* $a(x)$ (some sub-property or statistics of $x$) from a set of possible attributes $\mathcal{A}$. These attributes along with the labels form the set of possible groups $\mathcal{G} = \mathcal{A} \times \mathcal{Y}$ that each example can take. We denote an input $x$'s group label as $g(x) \in \mathcal{G}$. We then define the classification error of a predictor $f$ (w.r.t. a fixed implicit distribution) restricted to a group $g \in \mathcal{G}$ to be $L_g(f) := \mathbb{E}_{x,y|g(x)=g}[f(x) \neq y]$. The notion of *worst-group error* upper bounds the error of $f$ w.r.t. any group $L_{wg}(f) := \max_{g \in \mathcal{G}} L_g(f)$. Using this notation, the group-shift problem aims to discover a classifier in $\arg\min_{f \in \mathcal{F}} \{L_{wg}(f)\} = \arg\min_{f \in \mathcal{F}} \{\max_{g \in \mathcal{G}} L_g(f)\}$. We observe that the group-shift problem is just a special case of the domain generalization problem when $\mathcal{D}_{\text{test}}$ is the distribution consisting of only the points $(x, y)$ with $g(x)$ being restricted to the worst-group of $f$ in $\mathcal{G}$. Here, GDRO solves this objective by performing a minimax optimization procedure that alternates between the model's weight and relaxed weights on the groups.

**Spurious Correlations and Memorization.** As an example, consider the Waterbird dataset (Sagawa et al., 2020a), where it has been constructed by combining images of water/land birds from the CUB dataset (Welinder et al., 2010) with water/land backgrounds from the PLACE dataset (Zhou et al., 2017). The task is to distinguish whether an image of a bird is a waterbird or a landbird. In terms of our problem, the type of bird forms the labels $\mathcal{Y}$, and the backgrounds are set to be the attribute $\mathcal{A}$ for each type of bird. Altogether, these form four groups: $\mathcal{G} = \mathcal{Y} \times \mathcal{A} = \{\text{waterbird}, \text{landbird}\} \times \{\text{water}, \text{land}\}$.

---

**Algorithm 1** Classifier Retraining on Independent Splits (CROIS)

1: **Input:** Training data $D_L$ with group labels and training data without group labels $D_U$. Classifier retraining algorithm $\mathcal{R}$ (default to GDRO). Optional splitting parameter $p$ (default to 1).
2: *Obtain validation sets* by partitioning $D_L$ into $D'_L$ and $D_L^{(val)}$ and $D_U$ into $D'_U$ and $D_U^{(val)}$.
3: (Optional) *Add more unlabeled data* via split proportion $p$: Partition $D'_L$ into two parts, $D_1$ and $D_2$ such that $|D_1| = (1-p) \cdot |D'_L|$ and $|D_2| = p \cdot |D'_L|$. Set $D'_L \leftarrow D_2$ and $D'_U \leftarrow D'_U \cup D_1$.
4: *Obtain the initial model $f$* by running ERM on $D'_U$ and selecting the best model in terms of average accuracy on $D_L^{(val)} \cup D_U^{(val)}$.
5: *Perform classifier retraining $\mathcal{R}$* with feature extractor $f$ on $D'_L$ and then select the best model in terms of worst-group accuracy on $D_L^{(val)}$ as the final output.

---

This dataset is constructed so that the proportion of birds on matching backgrounds is significantly more than those of the mismatched backgrounds. This is so that the backgrounds could be spuriously correlated with the labels, as predicting the background alone would achieve a high average accuracy w.r.t. the train distribution already. As expected, for ERM trained models, the groups with the highest error are the minority groups where the background mismatches the type of the bird, suggesting that the model is actually predicting using the background instead of the bird. Furthermore, the fact that these high-capacity models achieve *zero* training error leads to the conclusion that these models not only utilize spurious features like the background to make its predictions, but also must have memorized the minority groups during its training process (Sagawa et al., 2020b). These problems are common when there is data imbalance (Feldman and Zhang, 2020). In the next section, we propose a method that attempts to circumvent these issues.

## 3  METHOD: CLASSIFIER RETRAINING ON INDEPENDENT SPLITS

Algorithm 1 presents an outline for our main method: *Classifier Retraining On Independent Splits* or *CROIS*. Given group-labeled data and group-unlabeled data, CROIS involves several steps: (1) organize the data into one *group-labeled* split $D'_L$ and one *group-unlabeled* split $D'_U$, (2) obtain an ERM trained feature extractor with the group-unlabeled split $D'_U$, and (3) perform robust classifier retraining with the group-labeled split $D'_L$, where classifier retraining refers to fine-tuning the final linear layer of a DNN. In the setting where group labels are limited (as in Section 4.1), $|D_L|$ is much smaller than $|D_U|$ and we do not need to set $p < 1$. There, we primarily concern with partitioning $D_L$ into $D'_L$ and $D_L^{(val)}$. On the other hand, when training group labels are available (as in Section 4.2) and $|D_U|$ is much smaller than $|D_L|$, the optional parameter $p$ in step 3 controls the size of $D'_U$ to obtain a feature extractor and the amount of group labels $D'_L$ actually used at train time.

**Motivation.** Inspired by previous works that have demonstrated the potential of simple ERM trained DNNs on a variety of OOD tasks (Gulrajani and Lopez-Paz, 2021; Rosenfeld et al., 2021) and long-tailed tasks (Kang et al., 2019), our method focuses on developing a simple method around ERM trained DNNs. As soon to be discussed, a *strength* of an ERM trained DNN is that there are convincing evidences that it contains *good features*. On the other hand, its *weaknesses* involve memorizing examples and using spurious features. CROIS is designed to alleviate these weaknesses while taking advantage of the good features of an ERM trained DNN.

**Models trained by ERM contains good features.** While not the first work to notice that ERM trained models contain good features, Kang et al. (2019) demonstrates this hypothesis through extensive experiments on several long-tailed vision datasets to investigate different learning strategies for obtaining a feature extractor as well as ways to fine-tune the classifier layer. There, an ERM trained feature extractor combined with a non-parametric method of rescaling[1] the classifier layer achieves (then) state-of-the-art results on all three datasets. A similar study (Menon et al., 2021) has been done to confirm this hypothesis on vision datasets in the group-shift setting. This suggests that one key to the data-imbalance problem (for which the group-shift problem almost always suffers) lies in correcting the classifier layer, which forms the first phase of CROIS. It is peculiar that rescaling the classifier works best whereas intuition suggests a data-dependent method like classifier retraining would be better. We hypothesize that this is related to the next issue of our discussion.

---

[1]Rescaling each row of the linear classifier using the row's norm to some power. See Kang et al. (2019).

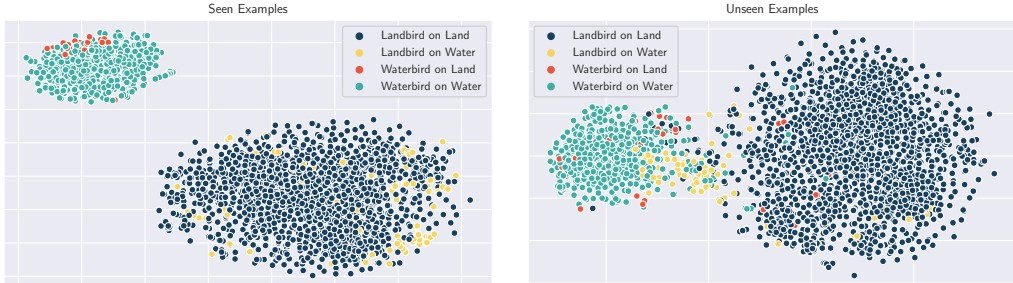

Figure 2: tSNE projection (Van der Maaten and Hinton, 2008) of the features of an ERM trained ResNet50 on *seen* (**left**) versus *unseen* (**right**) examples from Waterbird.[2] The features of the minority groups (orange and yellow) from the *unseen examples* appear better separated from the majority groups than that of the seen examples. Using unseen examples plays a major role in CROIS's ability to improve worst-group performance via robust classifier retraining.

**Memorization behavior of high capacity DNNs.** It has now been well known of high capacity DNNs' ability to memorize training examples (Zhang et al., 2017). In the group-shift setting, this behavior has been investigated by Sagawa et al. (2020b), which provides empirical and theoretical justifications for DNNs' memorization behavior of minority groups' training examples. This behavior of memorizing minority examples have also been observed in the broader framework of data imbalances (Feldman and Zhang, 2020). One way to circumvent memorization is to control the model's capacity by incorporating some combinations of high $\ell_2$ regularization, early stopping, and other correctional parameters as has been done in Sagawa et al. (2020a). However, the additional tuning required might be costly. We instead tackle this via *independent splits*.

**Circumventing memorization with independent splits.** Examples being memorized must inevitably impact their ability to be useful in subsequent usage. As memorized examples' (i.e. already correctly classified) loss must be low, their gradients contain little information to be of much use. Furthermore, the features of memorized examples might not be representative of their group during test time: Figure 2 presents a visualization of the features between seen versus unseen examples. Thus, combining this observation with the evidences for ERM trained DNNs containing good features, CROIS performs robust classifier retraining on *unseen* examples ($D'_L$ in Algorithm 1) in hope of learning a classifier that utilizes features more representative of examples during test time.

## 4 EXPERIMENTS

We conduct experiments in two settings: (1) where group labels are only available from the validation split of the datasets (as in Liu et al. (2021); Nam et al. (2022)); and (2) when a fraction of group labels is available from the training split and all group labels are available from the validation split.

**Setup.** We use a similar setup as in Liu et al. (2021) and Sagawa et al. (2020a). To demonstrate the ease of tuning of CROIS, unless noted otherwise (e.g. Table 2 and parameter $p$ in Table 3), we *fix* the hyperparameters of both the ERM and the robust classifier retraining phase, reusing standard parameters for ERM (see Appendix A for full hyperparameters and models details). Further results of CROIS with tuned hyperparameters are presented in Section C of the Appendix.

**Datasets.** We experiment on four datasets:

- **Waterbird** (Sagawa et al., 2020a). Combining the bird images from the CUB dataset (Welinder et al., 2010) with water or land backgrounds from the PLACES dataset (Zhou et al., 2017), the task is to classify whether an image contains a *landbird* or a *waterbird* without confounding with the background. There are 4795 total training examples, where the minority group (*waterbird, land background*) has only 56 examples. We report the weighted test *average accuracy* due to the skewed nature of the val and test sets to be consistent with Sagawa et al. (2020a).

- **CelebA** (Liu et al., 2015) is a popular image dataset of celebrity faces. The task is to classify the celebrity in the image is *blond* or *not blond*, with *male* or *not male* as the confounding attribute. There are 162770 total training examples, and the smallest group (*blond, male*) has 1387 examples.

---

[2]Half of the data is used to obtain a feature extractor while the other half is used to obtain the features of the unseen examples.

Table 1: Experimental results for the setting when only group labels from the validation split are used. Results for JTT and SSA are taken from Liu et al. (2021) and Nam et al. (2022), respectively. The numbers in parentheses denote one standard deviation from the mean across 3 random seeds. See Table E in Appendix 17 for comparison with additional baselines.

| Method | Waterbird | | CelebA | | MultiNLI | | CivilComments | |
|---|---|---|---|---|---|---|---|---|
| | Avg Acc | Wg Acc | Avg Acc | Wg Acc | Avg Acc | Wg Acc | Avg Acc | Wg Acc |
| JTT | 93.9 | 86.7 | 88.0 | 88.1 | 78.6 | 72.6 | 91.1 | 69.3 |
| SSA | 92.2 (0.87) | 89.0 (0.55) | 92.8 (0.11) | **89.8** (1.28) | 79.9 (0.87) | 76.6 (0.66) | 88.2 (1.95) | 69.9 (2.02) |
| CROIS | 92.1 (0.29) | **90.9** (0.12) | 91.6 (0.61) | 88.5 (0.87) | 81.4 (0.06) | **77.4** (1.21) | 90.6 (0.20) | **70.3** (0.34) |

- **MultiNLI** (Williams et al., 2017) is a natural language inference dataset for determining whether a sentence's hypothesis is *entailed* by, is *neutral* with, or *contradicts* its premise. The spurious attribute is the presence of negation words like *no, never*, or *nothing*. This task has 6 groups, with 206175 total training and 1521 in the minority group examples (*is entailed* and *contains negation*).

- **CivilComments-WILDS** (Borkan et al., 2019; Koh et al., 2021) is a natural language dataset where the task is to classify whether a sentence is *toxic* or *non-toxic*. There are 8 demographics – *male, female, white, black, LGBTQ, Muslim, Christian,* and *other religion*– forming 16 groups that *overlap*, because a comment can contain multiple demographics. Following Koh et al. (2021), we evaluate on all 16 groups but only use the attribute *black* along with the label in training. There is a total of 269038 training examples with 1045 minority examples from (*other religion*, *toxic*).

## 4.1 RESULT OF USING GROUP LABELS FROM THE VALIDATION SPLIT

**Setup.** In this section, we consider the setting where group labels $D_L$ are available only from the standard validation split, where these group labels can be used for training (Nam et al., 2022) and/or model selection (Liu et al., 2021). Here, the training split is treated as the group-unlabeled set $D_U$. Most methods in this setting employ some pseudo-labeling approach to generate pseudo-group-labels that are then used to train a *new* network via a robust algorithm like GDRO. On the other hand, CROIS simply uses half of $D_L$ for classifier retraining $D'_L$ and the other half for model selection $D_L^{(val)}$ and does not rely on pseudo-labeling. CROIS also reuses the initial model for the retraining phase, making CROIS closer to that of a single phase procedure with additional fine-tuning.

**Results.** In Table 1, we compare CROIS against JTT (Liu et al., 2021) and SSA (Nam et al., 2022), where we report the mean and one standard deviation of the Test Average (*Avg Acc*) and Worst-Group Accuracy (*Wg Acc*) across three random seeds. There, CROIS outperforms JTT on all 4 datasets and SSA on 3 datasets using default parameters. Note that, unlike CROIS and SSA, JTT only uses available group labels for model selection. However, JTT requires training a large number of models across two phases, which can be expensive. Furthermore, JTT's model selection can be quite sensitive (see Section 5.4 of Liu et al. (2021)). SSA alleviates this problem of JTT by more efficiently utilizing group-labels to infer pseudo-labeling. Finally, our method dispenses altogether with pseudo-labeling while still achieving competitive performance to JTT and SSA.

**Discussion.** We clarify the difference between JTT and CROIS. First, the initial ERM phase of JTT is for *inferring* pseudo-group-labels for the group-unlabeled data. This phase requires careful hyperparameter tuning and capacity control using the group-labeled validation set to accurately produce pseudo-group-labels (as noted in section 5.4 of Liu et al. (2021)). On the other hand, CROIS trains a *single* ERM model and simply retrains the last layer with any available group labels. Second, JTT's final performance is limited by GDRO's performance on the full network, which can be worsen by mislabeled pseudo-labels from the first phase. In contrast, we demonstrate in Section 4.2 that, by limiting GDRO to only the last layer, CROIS is competitive to full GDRO even when using only a fraction group labels and minimal tuning.

Compared with SSA, CROIS does not relying on pseudo labeling. In SSA, there is a phase that performs pseudo labeling and another phase that performs robust training with the inferred group labels. In the first phase, SSA trains a separate network that *predicts the group* rather than the class. By treating the pseudo-labeling problem as semi-supervised learning, SSA's pseudo-labeling capability is shown to improve upon JTT. Our results show that CROIS outperforms SSA on 3 out of 4 datasets while reusing default parameter.

Table 2: Worst-group test accuracy for partial group-labels from the validation split. Results for SSA and JTT are taken from Table 3 of Nam et al. (2022). The standard deviation is reported based on three independent runs. See Table 18 in Appendix E for comparison with additional baselines.

| % of group-labels from the validation split | CelebA | | | Waterbird | | |
|---|---|---|---|---|---|---|
| | 20% | 10% | 5% | 20% | 10% | 5% |
| JTT (Liu et al., 2021) | 81.1 | 81.1 | 82.2 | 84.0 | 86.9 | 76.0 |
| SSA (Nam et al., 2022) | 88.9 | **90.0** | 86.7 | 88.9 | **88.9** | 87.1 |
| CROIS's Wg Acc | **89.6** (0.4) | 87.6 (0.6) | **87.3** (1.0) | **90.4** (1.0) | 88.2 (0.9) | **87.8** (1.3) |
| CROIS's Avg Acc | 90.8 (0.2) | 91.6 (0.3) | 87.8 (1.6) | 92.4 (0.5) | 93.0 (0.7) | 88.7 (1.6) |

Table 3: Comparison between CROIS and GDRO. *NCRT* refers to naive classifier retraining i.e. when independent split is *not used* during the retraining phase. Results marked with [†] are taken from (Sagawa et al., 2020a). *For Waterbird, we omit the result for $p = 0.05$ due to the small dataset size and not being able to sample any minority-group example for robust retraining.

| | Waterbird | | CelebA | | MultiNLI | | CivilComments | |
|---|---|---|---|---|---|---|---|---|
| Method | Avg Acc | Wg Acc | Avg Acc | Wg Acc | Avg Acc | Wg Acc | Avg Acc | Wg Acc |
| ERM | 96.9 | 69.8 | 95.6 | 44.4 | 82.8 | 66.0 | 92.1 | 63.2 |
| GDRO | 93.2[†] | 86.0[†] | 91.8[†] | 88.3[†] | 81.4[†] | 77.7[†] | 89.6 (0.23) | 70.5 (2.10) |
| CROIS' $p$ – group-labeled fraction used for retraining (with $1 - p$ unlabeled fraction for the ERM phase) | | | | | | | | |
| 0.05 | * | * | 91.9 (0.50) | 88.9 (1.10) | 81.8 (0.15) | 73.8 (1.54) | 90.8 (0.40) | 63.3 (7.82) |
| 0.10 | 95.4 (1.10) | 83.5 (3.24) | 91.3 (0.36) | 90.3 (0.82) | 80.8 (0.51) | 75.3 (2.06) | 89.5 (1.81) | 68.7 (1.72) |
| 0.30 | 90.8 (0.35) | **89.6** (1.15) | 91.3 (0.44) | **90.6** (0.95) | 80.0 (0.31) | **77.9** (0.17) | 89.7 (0.33) | 68.6 (1.53) |
| 0.50 | 90.4 (0.95) | 89.5 (0.59) | 91.9 (0.35) | 88.2 (2.10) | 79.8 (0.26) | 74.4 (1.00) | 89.5 (0.70) | **71.0** (1.50) |
| NCRT | 96.5 | 75.2 | 93.9 | 69.2 | 82.3 | 67.9 | 90.3 | 67.6 |

**Reducing validation split size.** Following the setup in JTT and SSA, we vary the size of the validation split (20%, 10% and 5% of the original) to test whether our results still hold in these settings. We consider both the Waterbird and CelebA datasets. Note that for this setting, CROIS must be additionally *tuned* to account for the increased difficulty of the reduced group-labels quantity. Nevertheless, the smaller examples quantity along with just training the last layer makes the extra tuning not too expensive (details and setup in Section F). We present our results (along with error bars) in Table 2, where we find that CROIS outperforms JTT and SSA on various percentage levels.

## 4.2 RESULT OF USING PARTIAL TRAINING SPLIT GROUP LABELS

Next, we consider the setting where group labels are available from both the training split and the validation split. In contrast to Section 4.1, the standard validation split is used *only* for model selection $D_L^{(val)}$ and not for classifier retraining $D_L'$ here. We compare CROIS using some fraction of the training split's group labels against GDRO using *all* the group labels. Again, we fix parameters of CROIS to its standard ERM parameter to demonstrate its ease-of-tuning (see Appendix A).

**Setup.** We study CROIS with different amount of training group labels, determined by the parameter $p$. This means that $(1 - p)$ fraction of the training split is used to obtain a feature extractor in the first phase $D_U'$ (that uses no group label), and the rest $p$ fraction of group labels are used for robust classifier retraining $D_L'$. This setup allows examining the trade-off between the quality of the feature extractor versus the amount of data available to perform classifier retraining. Additionally, to demonstrate the importance of retraining with *unseen* examples, we experiment with robust classifier retraining using the same data from the first phase i.e. *without* independent splits – denoted as *NCRT*.

**The parameter $p$.** In practice, we expect that $|D_L|$ is a lot smaller than $|D_U|$, as in Section 4.1. There, Table 2 suggests that reasonable robust performance can be achieved with a small fraction of group labels. In this setup, however, since the amount of group labels is abundant ($D_U = \emptyset$ and $D_L$ is large), we treat $p$ as a tune-able parameter that controls the size of $D_U'$ and $D_L'$. Furthermore, using $p$ fraction of the available group labels simulates the process of obtaining group labels for a random fraction of the data if one is constraint by a budget for obtaining group labels.

**Results.** In Table 3, CROIS outperforms GDRO on both image datasets and yields competitive performance to GDRO on the two text datasets when using only a fraction of group labels and reusing default hyperparameters. Our result implies that comparable or even better robust performance than GDRO can be obtained by collecting group labels for roughly 30% of the available training data (modulo validation). One exception is in severe group imbalance case as in CivilComments (the minority group consists of only $0.4\%$ of the dataset). There, a higher fraction of group-labeled data is beneficial to obtain more minority-group examples. Hence, a more efficient sampling method to include more minority examples (e.g. filter by labels first) would be beneficial in practice. Finally, the results for NCRT also show the importance of using independent split for classifier retraining. [3]

**Trade-off between feature extractor and amount of group-labeled data for robust retraining.** From the results across the datasets, allocating more examples towards training the feature extractor (lower $p$) generally yields higher on-average accuracies. The worst-group error after classifier retraining has a more complex interaction with $p$, as it depends on both the quality of the feature extractor and the amount of group-labeled examples available to perform classifier retraining. While varying the proportion $p$ in our experiments gives a rough estimate of this tradeoff, we hypothesize that the availability of minority group examples is the most important for obtaining a robust classifier. We further support this intuition with an ablation study in Section D.2 where removing non-minority examples has an insignificant impact on the final group-robust performance.

**Alleviating GDRO's sensitivity towards model capacity.** GDRO's requirement for model capacity control via either $\ell_2$ regularization or early stopping is well noted in the literature (Sagawa et al., 2020a). In Table 13, we compare GDRO's and CROIS' sensitivity towards different $\ell_2$ regularization. While GDRO's performance on Waterbird is relatively uniform, GDRO is more sensitive to different $\ell_2$ settings on CelebA. When $\ell_2 = 1$ for CelebA, we see that GDRO fails altogether (see Figure 3). On the other hand, CROIS achieves consistent performance across different $\ell_2$ setting. CROIS controls the model capacity by limiting GDRO to only the last layer. This alleviates GDRO's tendency to overfit and simplifies parameter tuning (as in Figure 1 for Waterbird).

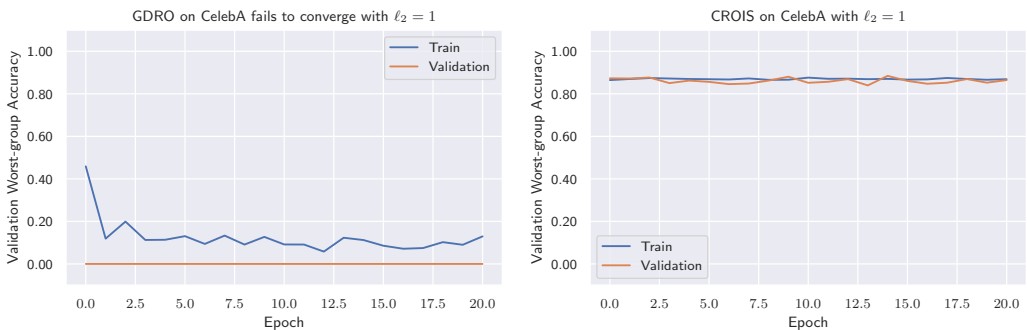

Figure 3: While GDRO often requires high $\ell_2$ regularization to avoid overfitting, setting $\ell_2$ too high might cause instability in GDRO's minimax optimization procedure (**left**).

### 4.3 ABLATION STUDIES AND DISCUSSIONS

**Obtaining a good feature extractor.** An ablation study on the effects of different validation accuracies and initial algorithm on the feature extractor's quality (measured by robust performance after classifier retraining) is presented in Section B. Similarly to previous works Kang et al. (2019), we find ERM providing the best features over reweighting or GDRO (of which both require group labels). We find that there's a positive correlation between validation average accuracy and features' quality. This then serves as a proxy for CROIS' model selection criterion in the first phase, which significantly simplifies parameter tuning over other two-phase methods in the group-shift setting.

---

[3]In Sagawa et al. (2020a), Group Adjustment (GA) (Cao et al., 2019) is observed to improve Waterbird's WG accuracy to 90.5%. We also notice an improvement when GA is incorporated to CROIS and obtain a $90.3\% \pm 0.62$ test WG accuracy. We also observe that, similarly to Sagawa et al. (2020a), GA only works for the synthetic dataset Waterbird but not for CelebA nor MultiNLI.

Table 4: Comparison between reweighting, subsampling, and GDRO as classifier retraining algorithms on top of the same feature extractor ($p = 0.30$).

| Retraining Method | Waterbird | | CelebA | | MultiNLI | | CivilComments | |
|---|---|---|---|---|---|---|---|---|
| | Avg Acc | Wg Acc | Avg Acc | Wg Acc | Avg Acc | Wg Acc | Avg Acc | Wg Acc |
| Reweighting | 95.2 | 87.1 | 92.1 | 85.0 | 78.9 | 67.0 | 88.3 | 56.4 |
| Subsampling | 95.8 | 81.1 | 91.6 | 86.1 | 78.6 | 64.0 | 91.8 | 59.3 |
| GDRO | 91.4 | **90.2** | 91.6 | **90.4** | 80.3 | **78.0** | 89.7 | **69.1** |

**GDRO is better than reweighting and subsampling for classifier retraining.** Table 4 contains results on using different classifier retraining methods. We observe that GDRO produces the best group-robust performance (since GDRO is designed for this setting after all). Reweighting and subsampling seem to be effective on the vision datasets, but fail to perform on the NLP datasets.

Table 5: Comparison between retraining with GDRO on the full network (*Full*) versus last layer (*LL*).

| GDRO $p = 0.30$ | Waterbird | | CelebA | | MultiNLI | | CivilComments | |
|---|---|---|---|---|---|---|---|---|
| | Avg Acc | Wg Acc | Avg Acc | Wg Acc | Avg Acc | Wg Acc | Avg Acc | Wg Acc |
| LL (CROIS) | 91.4 | **90.2** | 91.6 | **90.4** | 80.3 | **78.0** | 89.7 | 69.1 |
| Full | 90.5 | 79.8 | 91.6 | 78.3 | 80.8 | 75.1 | 90.4 | 69.1 |

**Classifier retraining outperforms full retraining with GDRO.** Classifier retraining plays a central role in our method. In Table 5, we compare between fine-tuning with GDRO on the full DNN versus just the last layer (LL) for an independent split of $p = 0.30$ . We see that GDRO on the LL is much better than full GDRO on most of the datasets (except CivilComments). However, the main difference is that while LL retraining (i.e. CROIS) requires little additional tuning, we must grid search for different regularization strength for GDRO when applied on the full network. CROIS can be shown to be quite robust to different parameter setting and regularization strength (details in Appendix C.1).

**Models learned with ERM contain good features.** The positive result for our decoupled training procedure provides another strong evidence for ERM trained models containing good features for the group-shift problem. While this is consistent with findings in the literatures on vision datasets (Kang et al., 2019; Menon et al., 2021), our work further provides some of the first evidences of this hypothesis in non-vision tasks, where the same result would not have been possible without independent split, as evident in the result for *Naive CRT* in Table 3.

**Simplified model selection.** The model selection criterion of picking the best average validation accuracy model simplifies hyperparameter tuning in comparison with other two-phases methods. This decision has been chosen mainly from the ablation experiments in Section B of the Appendix, where we observe that higher average validation accuracy generally suggests better features.

## 5 CONCLUSION AND FUTURE WORKS

In this paper, we propose Classifier Retraining on Independent Split (CROIS) as a simple method to reduce the amount of group annotations needed for improving worst-group performance as well as alleviate GDRO's requirement for careful control of model capacity. Our experimental results show the effectiveness of CROIS on four standard datasets across two settings and further provide evidences that ERM trained models contain good features for the group-shift problem.

**Future works.** The richness of ERM trained DNNs' features can potentially be useful towards solving the seemingly harder group-agnostic setting (where no group label is available) by allowing the practitioner to focus on obtaining a robust classifier given an ERM trained feature extractor, where we have shown that reasonable robustness can be achieved with relatively few group-labels, which makes the problem seem closer in reach. On a broader note, while most works in representation learning focus on producing good features (either with supervised, unsupervised, or self-supervised approaches), further examinations into different ways to perform classifier retraining in different settings (as in our work) could give a fuller picture to the features quality of different methods.

**Reproducibility statement.** We include our source code in the supplemental material. All implementation details and hyperparameters are detailed in Section A of the Appendix.

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

Table 6: Hyperparameters used in the experiments. The slash indicates the parameters used in the first phase (feature extractor) versus the second phase (classifier retraining).

|  | Waterbird | CelebA | MultiNLI | CivilComments |
|---|---|---|---|---|
| Learning Rate | $10^{-4}/10^{-4}$ | $10^{-4}/10^{-4}$ | $2 \times 10^{-5}/2 \times 10^{-5}$ | $10^{-5}/10^{-5}$ |
| $\ell_2$ Regularization | $10^{-4}/10^{-4}$ | $10^{-4}/10^{-4}$ | $0/0$ | $10^{-2}/0$ |
| Number of Epochs | 250/250 | 20/20 | 20/20 | 6/6 |

## A  EXPERIMENTAL DETAILS

### A.1  INFRASTRUCTURE

We performed our experiment on 2 PCs with one NVIDIA RTX3070 and one NVIDIA RTX3090. Our implementation is built on top of the code base from Liu et al. (2021). Experimental data was collected with the help of Weights and Biases (Biewald, 2020).

### A.2  MODELS

We use ResNet50 (He et al., 2016) with ImageNet initialization and batch-normalization for CelebA and Waterbird. We use pretrained BERT (Devlin et al., 2018) for MultiNLI and CivilComments. We use the original train-val-test split in all the datasets and report the *test* results. Cross-entropy is used as the base loss for all objectives. SGD with momentum $0.9$ is used for the vision datasets while the AdamW optimizer with dropout and a fixed linearly-decaying learning rate is used for BERT. We use a batch size of 16 for CivilComments and 32 for the rest of the datasets. We do *not* use any additional data augmentation or learning rate scheduler in our results.

### A.3  HYPERPARAMETERS

Table 6 contains the hyperparameters used in our experiments in Sections 4.2 and 4.1. Note that these are the standard parameters for obtaining an ERM model for these datasets as in previous works (Sagawa et al., 2020a; Liu et al., 2021). The only difference is that we train Waterbird and CelebA for slightly shorter epoch due to finding no further increase in validation accuracies after those epochs.

In our experiments, unless noted, we do not tune for any other hyperparameters. For the second phase of CivilComments, we do not use the default regularization but opt for $0$ since the linear layer already has low capacity. However, adding further regularization does not seem to have much of an effect as in section C.1.

## B  ABLATION STUDIES: OBTAINING A GOOD FEATURE EXTRACTOR

### B.1  IMPACT OF THE FEATURE EXTRACTOR'S ALGORITHMS

Here, we provide evidences that ERM trained models produce the best features for worst-group robustness. We conduct an experiment on Waterbird, where instead of using ERM to obtain a feature extractor, we perform GDRO and Reweighing instead on the first phase. The results are in table 7. While using reweighing or GDRO for the first phase defeats the purpose of reducing the amount of group-labels needed (whereas ERM doesn't need any), it is informative to examine the features alone.

Table 7: Effects of different methods for obtaining a feature extractor on test average accuracy and test worst-group accuracy (with ResNet50 on Waterbird).

| Feature extractor via | Test Avg Acc | Test Wg Acc |
|---|---|---|
| Reweighing | 90.1 | 88.8 |
| GDRO | 90.8 | 88.6 |
| ERM | 90.5 | **90.2** |

Here, we see that even though ERM does not use group-labels, it provides the best features for robust classifier retraining on independent split.

### B.2 IMPACT OF EARLY STOPPING AND VALIDATION ACCURACIES ON THE FEATURE EXTRACTOR

In this section, we present an ablation study of how different early stopping epoch (figure 4), average validation accuracy (figure 5 left), and worst-group accuracy (figure 5 right) of the initial ERM trained model (feature extractor) affect the GDRO retraining phase of CROIS. The results here are from performing CROIS with GDRO and $p = 0.30$ on Waterbird across a wide variety of epochs. Table 8 presents the full data generated for this section.

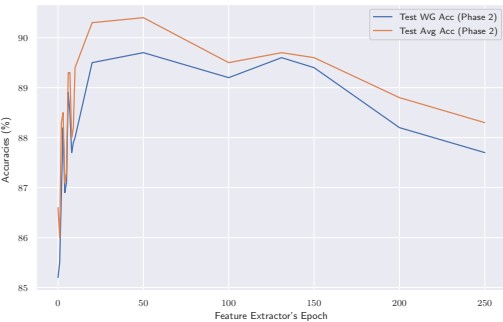

Figure 4: The effect of using different epochs for the feature extractor (phase 1) on classifier retraining's (phase 2) test accuracies.

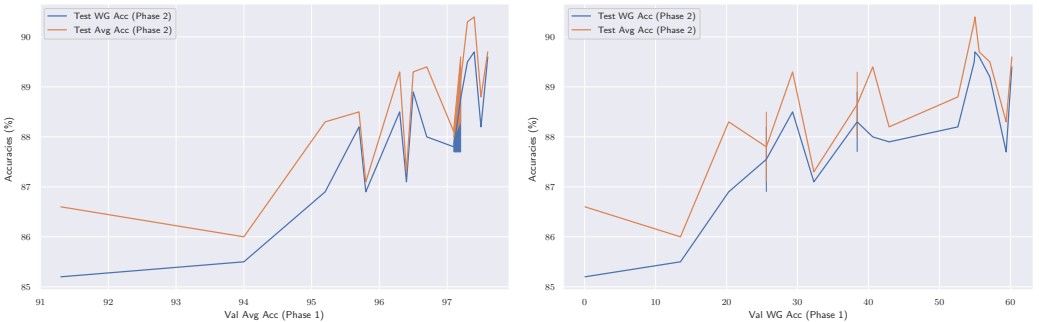

Figure 5: The effect of different *validation average accuracies* (**left**) and *validation worst-group accuracies* (**right**) from the feature extractor (phase 1) on classifier retraining's (phase 2) test accuracies.

## C HYPERPARAMETER TUNING: CROIS VS. GDRO

### C.1 FURTHER HYPERAMETER EXPLORATION ON CROIS

In our paper, we have demonstrated CROIS's effectiveness even with just using the same hyperparameters as to train an ERM model. In this section, we present results for further additional parameter tuning on the robust classifier retraining phase. These results provide empirical evidence for CROIS's potential as well as robustness to different hyperparameter setting.

$\ell_2$ **regularization.** We investigate whether *additional* regularization would be helpful to classifier retraining with GDRO on CelebA (Table 9) and Waterbird. (Table 10) We further examine the effects of regularization on CivilComments (Table 11) to support our choice in Section A. For these

Table 8: CROIS with GDRO ($p = 0.30$) on Waterbird. Average (*Avg Acc*) and Worst-group (*Wg Acc*) Accuracies for various epochs of the feature extractor ("Phase 1") and the corresponding test accuracies for classifier retraining ("Phase 2"). While training for longer epochs seem to help with average and worst-group accuracy for phase 2, the benefit is small. Hence, simply selecting the best validation average accuracy model, row Epoch 131 and denoted *BEST* here, yields good enough features that simplify our training procedure and model selection criteria.

| CROIS ($p = 0.3$) | Feature Extractor (Phase 1) | | Classifier Retraining (Phase 2) | | | |
|---|---|---|---|---|---|---|
| Phase 1 Epoch | Val Avg | Val WG | Val Avg | Val WG | Test Avg | Test WG |
| 0 | 91.3 | 0.05 | 87.8 | 85.6 | 86.6 | 85.2 |
| 1 | 94 | 13.5 | 88 | 87.6 | 86 | 85.5 |
| 2 | 95.2 | 20.3 | 89.6 | 88 | 88.3 | 86.9 |
| 3 | 95.7 | 25.6 | 90.1 | 88 | 88.5 | 88.2 |
| 4 | 95.8 | 25.6 | 88.9 | 88.2 | 87.1 | 86.9 |
| 5 | 96.4 | 32.3 | 89.2 | 88.7 | 87.3 | 87.1 |
| 6 | 96.5 | 38.4 | 90.5 | 88.7 | 89.3 | 88.9 |
| 7 | 96.3 | 29.3 | 90.6 | 88.7 | 89.3 | 88.5 |
| 8 | 97.1 | 38.4 | 90.1 | 89.3 | 88 | 87.7 |
| 9 | 97.1 | 42.9 | 90.4 | 89.9 | 88.2 | 87.9 |
| 10 | 96.7 | 40.6 | 91.2 | 90.2 | 89.4 | 88 |
| 20 | 97.3 | 54.9 | 91.4 | 91 | 90.3 | 89.5 |
| 50 | 97.4 | 55 | **91.5** | 91 | **90.4** | **89.7** |
| 100 | 97.2 | 57.1 | 90.5 | 90.2 | 89.5 | 89.2 |
| 131 (Best) | **97.6** | 55.6 | 90.9 | 90.2 | 89.7 | **89.6** |
| 150 | 97.2 | **60.2** | 90.7 | 90.2 | 89.6 | 89.4 |
| 200 | 97.5 | 52.6 | 91 | 90.2 | 88.8 | 88.2 |
| 250 | 97.2 | 59.4 | 91 | **90.4** | 88.3 | 87.7 |

experiments, we further run each setting acorss 3 random seeds to further investigate CROIS's sensitivity towards hyperparameters.

Table 9: Effects of $\ell_2$ regularization on classifier retraining with GDRO on CelebA.

| $\ell_2$ Reg. | $p = 0.50$ | | $p = 0.30$ | | $p = 0.10$ | |
|---|---|---|---|---|---|---|
| | Avg Acc | Wg Acc | Avg Acc | Wg Acc | Avg Acc | Wg Acc |
| 1 | 91.6 (0.06) | 87.4 (2.76) | 90.9 (1.13) | 88.4 (0.78) | 91.2 (0.35) | 90.0 (0.70) |
| $10^{-2}$ | 92.1 (0.32) | 87.6 (1.56) | **91.8** (0.21) | 87.5 (3.54) | 92.0 (0.44) | 88.3 (2.39) |
| $10^{-4}$ | 91.9 (0.35) | **88.2** (2.10) | 91.3 (0.44) | **90.6** (0.95) | 91.3 (0.36) | **90.3** (0.82) |
| 0 | **92.2** (0.25) | 86.8 (2.30) | 91.5 (0.07) | 87.8 (3.96) | **92.1** (0.47) | 88.1 (2.73) |

Table 10: Effects of $\ell_2$ regularization on classifier retraining with GDRO on Waterbird.

| $\ell_2$ Reg. | $p = 0.50$ | | $p = 0.30$ | | $p = 0.10$ | |
|---|---|---|---|---|---|---|
| | Avg Acc | Wg Acc | Avg Acc | Wg Acc | Avg Acc | Wg Acc |
| 1 | 89.6 (0.93) | 88.9 (1.37) | **91.7** (0.44) | **89.8** (0.68) | 94.5 (0.50) | 85.8 (0.15) |
| $10^{-2}$ | **90.4** (1.31) | 89.3 (0.61) | 90.6 (1.25) | 89.2 (1.31) | 95.1 (0.40) | 86.3 (0.83) |
| $10^{-4}$ | **90.4** (0.95) | **89.5** (0.59) | 90.8 (0.35) | 89.6 (1.15) | **95.4** (1.10) | 83.5 (3.24) |
| 0 | 89.8 (0.53) | 89.3 (0.70) | 90.6 (1.31) | 89.1 (1.16) | 95.1 (0.35) | **86.4** (0.90) |

Table 11: Effects of $\ell_2$ regularization on classifier retraining with GDRO on CivilComments.

| $\ell_2$ Reg. | $p = 0.50$ | | $p = 0.30$ | | $p = 0.10$ | |
|---|---|---|---|---|---|---|
| | Avg Acc | Wg Acc | Avg Acc | Wg Acc | Avg Acc | Wg Acc |
| 1 | 88.8 (1.30) | 70.0 (1.63) | 89.5 (0.35) | 66.4 (1.99) | 89.3 (1.69) | 68.9 (2.64) |
| $10^{-2}$ | 89.4 (0.99) | 70.6 (0.42) | 89.5 (0.29) | 68.5 (0.87) | 88.6 (1.70) | **70.2** (1.63) |
| 0 | **89.5** (0.70) | **71.0** (1.50) | **89.7** (0.33) | **68.6** (1.53) | **89.5** (1.81) | 68.7 (1.72) |

**Learning rate.** We examine the effects of different learning rates on CROIS on CelebA and Waterbird in Table 12. Lower learning rate seems to be more beneficial.

Table 12: Effects of varying learning rate on CROIS $p = 0.30$ on CelebA and Waterbird. We fix $\ell_2$ regularization to $10^{-4}$.

| | Learning rate | $10^{-5}$ | $10^{-4}$ | $10^{-3}$ | $10^{-2}$ | $10^{-1}$ |
|---|---|---|---|---|---|---|
| CelebA | Average accuracy | **92.2** | 91.4 | 91.2 | 90.1 | 91.2 |
| | Worst-group accuracy | 88.3 | 90 | **90.3** | 87.8 | 82.8 |
| Waterbird | Average accuracy | 89.7 | 90.6 | **94.2** | 93.2 | 93.9 |
| | Worst-group accuracy | **89.5** | 88.9 | 87.1 | 88.8 | 78.5 |

## C.2 SENSITIVITY TO MODEL CAPACITY: CROIS VERSUS GDRO

In this section, we present a comparison between CROIS and GDRO test performance with different $\ell_2$ regularization configuration on CelebA and Waterbird in Table 13. On CelebA, GDRO is quite sensitive to model capacity while it is less so on Waterbird. We note that GDRO fails to converge to a good stationary point when $\ell_2 = 1$ on CelebA (see Figure 3).

Table 13: Worst-group test accuracy for GDRO and CROIS ($p = 0.30$) with different $\ell_2$ regularization.

| $\ell_2$ reg. | CelebA | | | | | | Waterbird | | | | | |
|---|---|---|---|---|---|---|---|---|---|---|---|---|
| | 0 | $10^{-4}$ | $10^{-3}$ | $10^{-3}$ | $10^{-1}$ | 1 | 0 | $10^{-4}$ | $10^{-3}$ | $10^{-3}$ | $10^{-1}$ | 1 |
| GDRO | 81.7 | 81.7 | 81.7 | 83.9 | **87.8** | 0.00 | 86.8 | 86.8 | 86.8 | 86.8 | **87.1** | 86.5 |
| CROIS | 90.6 | **91.5** | 90.0 | 90.0 | 90.3 | 90.0 | 90.3 | **90.6** | 90.6 | 90.6 | 90.0 | 88.2 |

## D FURTHER STUDIES ON CLASSIFIER RETRAINING

### D.1 IMPACT OF ROBUST RETRAINING ON INDEPENDENT SPLITS

In this section, we examine how robust retraining affects the model's prediction of $D_U$ and $D_L$ before and after robust classifier retraining on independent split (with $p = 0.30$). Tables 14 and 15 show the accuracy on $D_U$ and $D_L$ for Waterbird and CelebA. The "Points changed" column indicates the number of points that the model changes prediction after robust retraining per group along with the total number of examples in that group (with percentage in parentheses). The worst-group is underlined in the tables.

Table 14: Model's prediction of $D_U$ and $D_L$ on Waterbird *before* robust retraining and *after* robust classifier retraining on independent split.

| Waterbird ($p = 0.3$) | Accuracy on $D_U$ | | | Accuracy on $D_L$ | | |
|---|---|---|---|---|---|---|
| | Before | After | Points changed | Before | After | Points changed |
| Avg Acc | 100 | 94.5 | 184/3356 (5.48%) | 96.4 | 89.6 | 162/1439 (11.3%) |
| Group 0 (73.0%) | 100 | 92.6 | 180/2430 (7.41%) | 99.6 | 88.2 | 122/1068 (11.4%) |
| Group 1 (3.84%) | 100 | 99.3 | 1/141 (7.09%) | 76.7 | 100 | 10/43 (23.3%) |
| Group 2 (1.17%) | 100 | 100 | 0/38 (0.00%) | 44.4 | 100 | 10/18 (55.6%) |
| Group 3 (22.0%) | 100 | 99.6 | 3/747 (0.40%) | 90.8 | 92.3 | 20/310 (6.45%) |

Table 15: Model's prediction of $D_U$ and $D_L$ on CelebA before and after robust classifier retraining on independent split. RR means after Robust Retraining.

| CelebA ($p = 0.30$) | Accuracy on $D_U$ | | | Accuracy on $D_L$ | | |
|---|---|---|---|---|---|---|
| | Before | After | Points changed | Before | After | Points changed |
| Avg Acc | 96.5 | 92.4 | 7537/113939 (6.61%) | 95.6 | 92.0 | 3274/48831 (11.3%) |
| Group 0 (43.7%) | 96.7 | 91.7 | 2609/50311 (5.19%) | 95.8 | 91.3 | 1011/21318 (4.74%) |
| Group 1 (41.4%) | 99.6 | 92.2 | 3489/46652 (7.48%) | 99.6 | 92.2 | 1493/20222 (7.38%) |
| Group 2 (14.1%) | 89.9 | 95.2 | 1000/16012 (6.25%) | 86.8 | 93.7 | 517/6868 (7.53%) |
| Group 3 (0.87%) | 46.3 | 91.8 | 439/964 (45.4%) | 35.5 | 95.3 | 253/423 (59.8%) |

Interestingly, after robust retraining, the worst-group almost always switches from the minority group to the majority group regardless of the data split.

## D.2 THE IMPORTANCE OF MINORITY EXAMPLES: SUBSAMPLED RETRAINING

As alluded to in Section 4.2, group-imbalance seems to play an important role in the robust performance of CROIS. To further demonstrate this point, we consider how CROIS performs when the second phase is subsampled versus when it is allowed additional non-minority examples.

Table 16: Performance of CROIS when retraining is on a subsampled split versus the full split. Subsampling the split doesn't seem to impact CROIS's performance, indicating the importance of the availability of minority examples.

| Dataset (minority group size, fraction) | Subsampled retraining | | Full retraining | |
|---|---|---|---|---|
| | Avg Acc | Wg Acc | Avg Acc | Wg Acc |
| CelebA ($n = 423$, 0.87%) | 90.6 | 89.7 | 91.9 | 88.9 |
| Waterbird ($n = 18$, 1.2%) | 91.3 | 87.6 | 90.3 | 88.9 |

As the result in Table 16 shows, there isn't a significant difference between the two sampling strategies, suggesting that the availability of minority group examples plays an important role in robust classifier retraining.

## E ADDITIONAL COMPARISON

We provide additional baselines for comparison in the Tables below.

Table 17: Experimental results for the setting when only group labels from the validation set are used. Results for C-DRO (CVaR DRO), LfF, EIIL, JTT and SSA are from Nam et al. (2022). Results for UMIX are from Han et al. (2022). Results for CnC are from Zhang et al. (2022). The numbers in parentheses denote one standard deviation from the mean across 3 random seeds.

| Method | Waterbird | | CelebA | | MultiNLI | | CivilComments | |
|---|---|---|---|---|---|---|---|---|
| | Avg Acc | Wg Acc | Avg Acc | Wg Acc | Avg Acc | Wg Acc | Avg Acc | Wg Acc |
| C-DRO | 96.0 | 75.9 | 82.5 | 64.4 | 82.0 | 68.0 | 92.5 | 60.5 |
| LfF | 91.2 | 78.0 | 85.1 | 77.2 | 80.8 | 70.2 | 92.5 | 58.8 |
| EIIL | 91.2 | 78.0 | 85.1 | 77.2 | 80.8 | 70.2 | 92.5 | 58.8 |
| JTT | 93.9 | 86.7 | 88.0 | 88.1 | 78.6 | 72.6 | 91.1 | 69.3 |
| UMIX | 93.0 (0.5) | 90.0 (1.1) | 90.1 (0.4) | 85.3 (4.1) | N/A | N/A | 90.6 (0.4) | 70.1 (0.9) |
| CnC | 90.9 (0.1) | 88.5 (0.3) | 89.9 (0.9) | 88.8 (0.9) | N/A | N/A | 81.7 (0.5) | 68.9 (2.1) |
| SSA | 92.2 (0.87) | 89.0 (0.55) | 92.8 (0.11) | **89.8** (1.28) | 79.9 (0.87) | 76.6 (0.66) | 88.2 (1.95) | 69.9 (2.02) |
| CROIS | 92.1 (0.29) | **90.9** (0.12) | 91.6 (0.61) | 88.5 (0.87) | 81.4 (0.06) | **77.4** (1.21) | 90.6 (0.20) | **70.3** (0.34) |

Table 18: Additional comparison where group labels from the training set are available. Results for LISA are taken from Yao et al. (2022). Results for CAMEL are taken from Goel et al. (2020).

| Method | Waterbird | | CelebA | | MultiNLI | | CivilComments | |
|---|---|---|---|---|---|---|---|---|
| | Avg Acc | Wg Acc | Avg Acc | Wg Acc | Avg Acc | Wg Acc | Avg Acc | Wg Acc |
| ERM | 96.9 | 69.8 | 95.6 | 44.4 | 82.8 | 66.0 | 92.1 | 63.2 |
| GDRO | 93.2$^\dagger$ | 86.0$^\dagger$ | 91.8$^\dagger$ | 88.3$^\dagger$ | 81.4$^\dagger$ | 77.7$^\dagger$ | 89.6 (0.23) | 70.5 (2.10) |
| LISA | 91.8 (0.3) | 89.2 (0.6) | 92.4 (0.4) | 89.3 (1.1) | N/A | N/A | 89.2 (0.9) | **72.6** (0.1) |
| CAMEL | 90.9 (0.9) | 89.1 (0.4) | N/A | N/A | N/A | N/A | N/A | N/A |
| CROIS' $p$ – group-labeled fraction used for retraining (with $1 - p$ unlabeled fraction for the ERM phase) | | | | | | | | |
| 0.10 | 95.4 (1.10) | 83.5 (3.24) | 91.3 (0.36) | 90.3 (0.82) | 80.8 (0.51) | 75.3 (2.06) | 89.5 (1.81) | 68.7 (1.72) |
| 0.30 | 90.8 (0.35) | **89.6** (1.15) | 91.3 (0.44) | **90.6** (0.95) | 80.0 (0.31) | **77.9** (0.17) | 89.7 (0.33) | 68.6 (1.53) |

### E.1 ADDITIONAL BASELINES FOR GROUP LABELS FROM ONLY THE VALIDATION SET

**Baselines.** In Table 17, we compare CROIS against JTT (Liu et al., 2021) and SSA (Nam et al., 2022), as well as additional baselines like CVaR DRO (Levy et al., 2020), LfF (Nam et al., 2020), EIIL (Creager et al., 2021), CnC (Zhang et al., 2022) and UMIX (Han et al., 2022). There, we report the mean and one standard deviation of the Test Average (*Avg Acc*) and Worst-Group Accuracy (*Wg Acc*) across three random seeds.

### E.2 ADDITIONAL BASELINES FOR GROUP LABELS FROM TRAINING SET

In the setting where group labels are available from the training set, we compare our method against additional baselines like LISA (Yao et al., 2022).

## F FRACTION OF THE VALIDATION SET IMPLEMENTATION DETAILS FROM SECTION 4.1

Following the setup in Section 4.1 and the setup as in Liu et al. (2021); Nam et al. (2022), we further reduce the validation set to only a small fraction, 5%, 10%, and 20%. We investigate CROIS's performance in this very few group-labels setting across CelebA and Waterbird in Section 4.1. We note that the highly reduced sample size poses new challenge and makes it harder to simply reuse the default parameters.

- **Tuning $\ell_2$ regularization:** When using so little data, overfitting can become a bigger problem, even when just training a low-capacity linear classifier. Hence, we tune for higher values for $\ell_2$ regularization across $\{10^{-4}, 10^{-2}, 1, 10\}$.

- **Tuning learning rate:** We also tune the learning rate across $\{10^{-5}, 10^{-4}, 10^{-3}, 10^{-2}\}$ instead of simply reusing default parameters.

- **The use of group-labels and model selection:** Since the amount of examples for classifier retraining is now significantly reduced, it might be wasteful to further split our available group-labels for validation. Instead, we use all the available group-labels for robust classifier retraining and perform model selection in the second phase via the *train worst-group accuracy*. The low capacity linear layer and higher $\ell_2$ regularization allows us to avoid overfitting when performing model selection this way. The feature extractor from the first phase is selected via the best average accuracy on the full group-unlabeled validation set.

- **Smaller batch size:** Since GDRO requires group-balanced sampling, a batch size greater than the number of examples in a certain group would cause duplicate sampling of the minority-group examples in the same step, artificially increasing the weight for that group. We further tune for batch sizes across a grid of powers of 2 less than the smallest group or the default batch size (e.g. we search across $\{4, 8, 16\}$ if the size of the smallest group is 17).

In Table 2, we present the results for CROIS with the above modifications and compare it to CROIS and JTT. There, robust retraining for CelebA is performed with an $\ell_2$ regularization of 0.1, batch size of 8 and learning rate $10^{-5}$. For Waterbird, we found that batch size 8, weight decay 1, and learning rate $10^{-5}$ are best for 20% and 10% reduction. For 5% reduction, we further reduce the batch size to 4 (since the minority group only has 7 examples) and increase the weight decay to 10.

The results shows that CROIS maintains its robustness performance even at greatly reduced group-labels. This implies that even a few (minority) examples can help debias the final layer classifier with proper configurations.

## G    IMPLICATIONS AND TAKEAWAYS

**Comparison to standard pretraining and finetuning**    As mentioned in the related works section as well as during the discussion of the motivation for our method, pretraining and then finetuning is a now well-known and established strategy in many domains. Our work differs the most significantly from this standard strategy through:

1. The use of independent splits: Traditional pretraining and finetuning reuses the dataset for both phase with the possibility of additional labels (contrastive learning, long-tail learning, etc.). We have demonstrated through extensive experiments the importance of independent split for classifier retraining to work well for the group robust setting.

2. The use of a group robust algorithm for finetuning: We mainly utilize GDRO for our finetuning phase while most other works in finetuning make use of simpler strategy for finetuning like reweighing or subsampling. We demonstrated in our experiments that GDRO yields the best robust performance over other more common methods.

Our work provides evidences that the features of ERM trained DNNs are rich enough to solve the group-shift problem (when an abundant amount of group labels is available to retrain the classifier) and one of the major reasons for poor worst-group performance of an ERM trained DNN is within its classifier layer. We then further demonstrate that even a few group labels can sufficiently "fix" the classifier to achieve better group-robust performance.

This knowledge can potentially be useful towards solving the seemingly much harder group-agnostic setting (where no group label is available) by allowing the practitioner to focus on obtaining a robust classifier given an ERM trained feature extractor. Our experiments further show that reasonable robustness can be achieved with relatively few group-labels (that are not used to obtain the feature extractor), which makes the problem seem closer in reach.

