# OpenReview forum: "Improved Group Robustness via Classifier Retraining on Independent Splits"
_ICLR.cc/2023/Conference — Submitted to ICLR 2023_

### Official Review · Reviewer_ysYp · 2022-10-23

**Confidence:** 4
**Clarity, Quality, Novelty And Reproducibility:** This paper is well written and easy t…
**Correctness:** 3
**Technical Novelty And Significance:** 3
**Empirical Novelty And Significance:** 3
**Recommendation:** 5

**Strength And Weaknesses:**

Advantages.

This paper is well written and easy to follow. Although the method of the paper is too straightforward, the author explains its principle very clearly.

Weaknesses.

However, I still have the following concerns:
1. The comparison experiments in Table 1 is unfair. Since the proposed method uses the data from the validation set for training.
2. The paper does not conduct discussion and compare with state-of-the-art methods, e.g., LISA[1], UMIX[2], CnC[3], CGD[4].

[1] Improving out-of-distribution robustness via selective augmentation.

[2] UMIX: Improving Importance Weighting for Subpopulation Shift via Uncertainty-Aware Mixup

[3] Correct-n-Contrast: A Contrastive Approach for Improving Robustness to Spurious Correlations

[4] Improving out-of-distribution robustness via selective augmentation.

**Summary Of The Paper:**

This paper proposes a very simple method termed classifier retraining on independent splits to improve the subpopulation shift robustness of the model. The main algorithm of the paper is clearly shown in Algorithm 1 of the paper.

**Summary Of The Review:**

This paper proposes a very simple and effective method. But some experiments are unfair.

---

> ### Author Response · Authors · 2022-11-10
> **Author response**
>
> We thank the reviewer for the comments! We would like to address the reviewer's concerns below:
>
> > 1. The comparison experiments in Table 1 is unfair. Since the proposed method uses the data from the validation set for training.
>
> Our setup in Tables 1 and 2 is identical to SSA (and almost identical to JTT). We would like to clarify that SSA extensively utilizes group labels from the validation set for both training a group labeling network and model selection. For JTT, while group labels are only used for model selection, their method requires model selection across a large number of models across two phases for both pseudo labeling and robust training, which can be expensive. Furthermore, JTT’s model selection criterion is quite sensitive (see Section 5.4 of [2]). SSA alleviates this problem of JTT by utilizing the group labels more efficiently to infer pseudo-group-labels. Finally, our method, CROIS, dispenses with pseudo labelling altogether while still achieving competitive performance to JTT and SSA. Most importantly, CROIS, JTT, and SSA all have the same access to the same amount of group labels in this setting.
>
> [1] Liu et al. Just Train Twice: Improving Group Robustness without Training Group Information. 2021.
>
> > 2. The paper does not conduct discussion and compare with state-of-the-art methods, e.g., LISA[1], UMIX[2], CnC[3], CGD[4].
>
> Thank you for the references! Some of these are very new and are published after our submission (e.g. UMIX [2] will be published in Neurips22 which has not happened yet). Furthermore, CnC [3] requires an expensive contrastive learning step in the second phase, and the contrastive learning step involves many hyperparameters. On the other hand, the approach in our work is simpler and requires fewer hyperparameters. Additionally, LISA[1] presents a technique that is more orthogonal to our work by utilizing data augmentation in the setting when abundant training group labels are available. However, we agree that a comprehensive comparison is important. We include 2 tables below for comparison and will update our paper with more baseline when more space is allotted.
>
> CROIS, CnC, and UMIX can be compared in the setting when only group labels from the validation set are available.
> | Avg/Wg | Waterbird | CelebA | CivilComments | MNLI |
> |---|---|---|---|---|
> | CnC [3] | 90.9 (0.1/88.5 (0.3) | 89.9 (0.9)/**88.8 (0.9)** | 81.7 (0.5)/68.9 (2.1) | N/A |
> | UMIX [2] | 93.0 (0.5)/90.0 (1.1) | 90.1 (0.4)/85.3 (4.1) | 90.6 (0.4)/70.1 (0.9) | N/A |
> | CROIS | 92.1 (0.3)/**90.9 (0.1)** | 91.6 (0.6)/**88.5 (0.9)** | 90.6 (0.2)/**70.3 (0.3)** | 81.4 (0.1)/**77.4 (1.2)** |
>
> CROIS and LISA can be compared in the setting when training group labels are available. Note that CROIS uses only 30%-10% of the training group labels versus LISA using 100% of the training group labels.
>
> | Avg/Wg | % of group labels use | Waterbird | CelebA | CivilComments | MNLI |
> |---|---|---|---|---|---|
> | LISA [1] | 100% | 91.8 (0.3)/89.2 (0.6) | 92.4 (0.4)/89.3 (1.1) | 89.2 (0.9)/**72.6 (0.1)** | N/A |
> | CROIS  | 30% | 90.8 (0.4)/**89.6 (1.15)** | 91.3 (0.4)/**90.6 (0.95)** | 89.7 (0.3)/68.6 (1.5) | 80.0 (0.3)/**77.9 (0.2)** |
> | CROIS  | 10% | 95.4 (1.1)/83.5 (3.2) | 91.3 (0.4)/90.3 (0.8) | 89.5 (1.8)/68.7 (1.7) | 80.8 (0.5)/75.3 (2.1) |
>
>
> [1] Improving out-of-distribution robustness via selective augmentation.
>
> [2] UMIX: Improving Importance Weighting for Subpopulation Shift via Uncertainty-Aware Mixup
>
> [3] Correct-n-Contrast: A Contrastive Approach for Improving Robustness to Spurious Correlations
>
> [4] Improving out-of-distribution robustness via selective augmentation.

---

> > ### Comment · Reviewer_ysYp · 2022-11-28
> > **Response**
> >
> > Thanks for your responses to me.
> > I read this paper and related work carefully again. I think the main weakness of this work is the innovation of the method and the lack of contribution.
> > The phenomenon mentioned in this paper has also been found in other papers [1][2]. Meanwhile, the method proposed in this paper has also been proposed in previous work[2]. The method in this paper appears to be a small improvement over the method proposed in [2].
> > So I keep my score fixed.
> > [1] Decoupling representation and classifier for long-tailed recognition.
> > [2] Overparameterisation and worst-case generalisation: friend or foe?

---

> > > ### Author Response · Authors · 2022-11-29
> > > **Author Reponse**
> > >
> > > Thanks again for the reviewer’s time and carefully checking our paper -- we appreciate it! Regarding the reviewer’s new comments about the weaknesses of this work, we respectfully highlight a few points below.
> > >
> > > **Innovation of the method.** First, compared with the approach of Kang et al. ’19 and Menon et al. ’21, our method shares a classifier retraining component while introducing a novel sample-splitting procedure. Notice that if we were to run the experiments without the sampling splitting procedure, the results would have become much worse— see reported results in Table 3. In particular, the NCRT line only uses classifier retraining without sampling splitting. Notice that for the Waterbirds dataset, our method outperforms this line by over 14% of worst-group accuracy! This ablation study suggests that incorporating sample-splitting with independent splits is essential (and not just a small improvement over Menon et al.) for the final performance of our method.
> > >
> > > Second, compared with the experimental results of Menon et al. ‘21, we have additionally provided evaluations for NLP tasks, including CivilComments and MultiNLI.
> > >
> > > **Regarding contributions.** At a technical level, our method is competitive in the full group labels setting with GroupDRO and in the partial group labels setting with SSA. This allows us to understand the relative performance compared with procedures that require inferring pseudo-group labels, which would not be possible with the insights from Menon et al.
> > >
> > > At a conceptual level, in a field where lots of methods have been designed, each involving lots of hyperparameters, we are surprised by our empirical findings that a simple approach works well for spurious correlation datasets such as Waterbirds and CelebA. Our method may also help study long-tailed generalization but this is beyond our scope. This would be an interesting research question for future work. We hope our work inspires future works to consider the design of simple methods for tackling spurious correlations.
> > >
> > > We are happy to continue the discussion and clarify if you have any more questions/comments. Thanks again for your time!

---

### Official Review · Reviewer_2dWG · 2022-10-26

**Confidence:** 4
**Correctness:** 3
**Technical Novelty And Significance:** 3
**Empirical Novelty And Significance:** 3
**Recommendation:** 6

**Clarity, Quality, Novelty And Reproducibility:**

The paper is well presented (except for the choice of a key hparam of CR data size, see weakness #2 above). The proposed method is a simple, novel and interesting idea that is orthogonal to the usual pseudo-group-labeling approach taken by other existing methods like JTT. I think most readers will find it interesting. The pseudo code is clear and simple enough for interested readers to reproduce the work.

**Strength And Weaknesses:**

Strengths:

The proposed method is a simple, novel and interesting idea that is orthogonal to the usual pseudo-group-labeling approach taken by other existing methods like JTT.

Weaknesses/comments:

1. The relationship of the CROIS performance to the data size for retraining (group-labeled fraction in training set for CR - Table 3, or reduced val size for CR - Table 2) seems quite inconclusive, as also pointed out by the authors in Sec 4.2 (Page 8). In particular when one cross compares Table 2 and 3, I assume for instance 30% of training size (Table 3) is comparable to 100% of val size (Table 2) for CelebA, which happens to be the sweet CR size for this dataset? IMHO p is the (only) key parameter in Algorithm 1, and it would be good to clarify in the paper, for instance, what's the rule of thumb of choosing p? Would it make sense to consider the CR data size (or p) as a tunable hparam? If group-unlabeled size >> group-labeled size, then should one use 50/50 group-labeled in D_L/D_val or tuning the ratio is still needed?
2. Empirical results are somewhat strong, on par with (but not significantly outperforming) SOTA (e.g. Table 1). Many other baselines (Sec 1.1) should be added to the results (e.g. Table 1) for comparison. Besides, it would be good to include error bars in Table 2 to show that error bar should become much larger when val size becomes smaller despite the "relatively small" drop in mean performance.
3. In Sec 3, the authors claim that independent splits are done to avoid the feature extractors memorizing spurious correlations with class labels. But many modern DL models rely on unsupervised/contrastive/label-free training of feature extractors. For instance, one can continue the pretraining of BERT on the in-domain data on the two language tasks (https://arxiv.org/abs/2004.10964). Perhaps then independent splits still won't be required and more group labels can be used for classifier retraining (in case of limited group annotations)?
4. The abstract/intro clearly separates the two use cases: a) full training and b) val group annotation. Algorithm 1 (Sec 3) does not make such distinction where D_L is listed as an input. It only becomes clear how Alg 1 is applied to case b) in Sec 4.1 where val set is actually split 50/50 into D_L and D_val. For consistency, could consider clarifying this earlier in the paper to avoid confusion.

**Summary Of The Paper:**

The authors proposed a novel approach to improve worst-group performance, CROIS, that works well with reduced/limited group annotations on training/val set. The idea is simple yet effective: obtain good feature extractors on group-unlabeled data with ERM loss, and only retrain the last layer on group-labeled data with GDRO loss. Empirical experiments and abundant ablations are performed to support the method.

**Summary Of The Review:**

Though the empirical results do not outperform SOTA, the paper still presents a novel and interesting idea in worst-group improvement that is quite different from usual approaches in this area. A key question regarding the group-labeled size for classifier retraining can be better explained for other researchers to build upon the work and/or practitioners to apply the proposed method.

---

> ### Author Response · Authors · 2022-11-10
> **Author response**
>
> We thank the reviewer for the positive comments and valuable feedback! We would like to address the reviewer's comments below.
>
> > it would be good to clarify in the paper, for instance, what's the rule of thumb of choosing p? Would it make sense to consider the CR data size (or p) as a tunable hparam?
>
> From Section 4.2 or the full group labels setup, our experimental results suggest that using 30% of the dataset for classifier retraining generally achieves the right balance if one has to make a tradeoff between allocating data for obtaining feature extractor versus saving data for classifier retraining. If one has an abundant amount of group labels, as in Section 4.2, one could consider $p$ a tunable hyperparameter. We will make this distinction clearer in our revised draft.
>
> For the limited group labels setup in Section 4.1, since the group-unlabeled data for the feature extractor is fixed, our main decision is simply in how we utilize the portion of group-labeled data. As expected, the more group-labels that are available the better performance that we can achieve (as in Table 2). It is only when we have to make a tradeoff between allocating more group labels for classifier retraining versus data for obtaining a feature extractor (as in Table 3) that balancing the tradeoff via $p$ seems to be important. The setup in Section 4.2 hopes to study such a tradeoff, but we expect that the limited group-labels setup from Section 4.1 might be closer to a practical scenario.
>
> > If group-unlabeled size >> group-labeled size, then should one use 50/50 group-labeled in D_L/D_val or tuning the ratio is still needed?
>
> Our experimental results from Table 1 in Section 4.1 suggest that if the group-labeled size is sufficiently large (say 100% the size of the val set), then a 50/50 split seems to be sufficient (where we chose such a split proportion for simplicity). On the other hand, the results from the very small group-labeled size setting as in Table 2 require a more careful tuning of not only the amount of group labels for D_L/D_val but also the learning rate, regularization, etc. to account for the very small amount of group labels (detailed in Appendix E).
>
> > Many other baselines (Sec 1.1) should be added to the results (e.g. Table 1) for comparison. Besides, it would be good to include error bars in Table 2 to show that error bar should become much larger when val size becomes smaller despite the "relatively small" drop in mean performance.
>
> Thank you for the suggestions! We have performed additional runs to show error bars for Table 2. We will be adding these to Table 2 in our revision.
>
> |  | CelebA |  |  | Waterbird |  |  |
> |---|---|---|---|---|---|---|
> | % of val. | 20% | 10% | 5% | 20% | 10% | 5% |
> | JTT | 81.1 | 81.1 | 82.2 | 84.0 | 86.9 | 76.0 |
> | SSA | 88.9 | **90.0** | 86.7 | 88.9 | **88.9** | 87.1 |
> | CROIS (wg acc) | **89.6 (0.4)** | 87.6 (0.6) | **87.3 (1.0)** | **90.4 (1.0)** | 88.2 (0.9) | **87.8 (1.3)** |
> | CROIS (avg acc) | 90.8 (0.2) | 91.6 (0.3) | 87.8 (1.6) | 92.4 (0.5) | 93.0 (0.7) | 88.7 (1.6) |
>
> Due to space constraint, we only included the strongest methods of Section 4.2 in our original draft. However, we agree that it is important to include more baselines, so we will revise our draft with additional baselines (Lff, CVARDRO, and EIIL) in Table 1 when more space is allotted.
>
> > many modern DL models rely on unsupervised/contrastive/label-free training of feature extractors. [...] Perhaps then independent splits still won't be required and more group labels can be used for classifier retraining (in case of limited group annotations)?
>
> Thank you for pointing out this interesting work and direction! An in-depth investigation into the effects of different methods to obtain a feature extractor on group robustness seems of great interest for this problem. One of our main hypotheses for performing an independent split is to avoid biasing the classifier retraining phase with the (likely) easily memorized minority examples. Hence, if the pretraining task does not bias the model towards the label of the task, it seems possible that we won’t need an independent split and more group labels (perhaps all?) can be used for robust classifier retraining.
>
> > The abstract/intro clearly separates the two use cases: a) full training and b) val group annotation. Algorithm 1 (Sec 3) does not make such distinction where D_L is listed as an input. It only becomes clear how Alg 1 is applied to case b) in Sec 4.1 where val set is actually split 50/50 into D_L and D_val. For consistency, could consider clarifying this earlier in the paper to avoid confusion.
>
> We apologize for the confusion, and thank you for the suggestion. We will clarify the different settings for our algorithm along with adding relevant examples in our revised draft.

---

### Official Review · Reviewer_X9ti · 2022-11-01

**Confidence:** 4
**Correctness:** 3
**Technical Novelty And Significance:** 3
**Empirical Novelty And Significance:** 3
**Recommendation:** 5

**Clarity, Quality, Novelty And Reproducibility:**

**Clarity**: The paper is well written, but the “Experiments” section is not always very clear (e.g. the reported results are the average or the best values? What are the authors varying to create 3 runs? Why is the std. not included in all the Tables?)

**Quality**: As highlighted in the “weaknesses” the quality of the experiments can be improved.

**Novelty**: Although the single components exist in prior works, the idea of combining the two stages is somewhat new.

**Reproducibility**: The source code is provided and the hyperparameters are detailed in the Appendix.


**Strength And Weaknesses:**

**Strength**:

The paper is well written, with simple and relevant notation. CROIS requires fewer labelled data than the GDRO baseline considered, and it does not rely on pseudo-labelling as JTT. The authors perform a good ablation study on different components of their proposed method (e.g. sampling with or without replacement, i.e. using independent splits or not, first and second stage different training strategies, robustness towards the hyperparameters tuning).

**Weaknesses**:

Fully training group labels baselines could include more recent methods such as SGDRO (non-flat version of GDRO proposed in Goel et al.).
There is a misleading sentence on page 3 when describing the Waterbirds dataset: “The bird images are then modified with either a water or land background.” There is no “modification” involved, in the dataset the birds are already placed either on water or land background (this is a minor note, but should be revisited).
The main weaknesses are all in the “Experiments” section. 1) Although the hyperparameters are fixed, some ablation is still required (for example the regularisation in the second stage for the CivilComments dataset). 2) In Tables 1 and 2 it is not clear what the results are, is it correct to assume they are the mean and std. performances over three different initialization seeds? 3) The results in Tables 1 and 2 are not correct, CROIS is using the validation set for training in the second stage, not only for model selection. For the datasets considered, the validation set has the same distribution as the test set, it does not seem correct to talk about “group shift” anymore. 4) Table 2 should also include the average test accuracy to show the trade-off between average and worst-case performances when varying the size of the validation set. Also, why is the std. not reported here? 5) Table 3 shows promising results, but the best fraction “p” is a hyperparameter that should be tuned using the worst-group performance on the validation set, like the regularization term for GDRO.  Its behaviour is not linear (e.g. the more the better) nor consistent across datasets. For example, in both text datasets, only one fraction produces better results than the GDRO baseline. Here, I disagree with the statement “In practice, p is not a parameter to choose (there’s no reason to throw away group labels) but rather is limited by the resources available to obtain group labels”: using p=0.5 is often worse than using p=0.3. 6) To obtain robust results, it would have been better to evaluate the methods across different splits of train-val-test, not simply different initialisation seeds.


**Summary Of The Paper:**

The paper proposes a simple method called CROIS for learning a model that can be applied in group-shift settings, i.e. cases where the performance of the group that is underrepresented in the training set is low when there is a distribution shift.
CROIS is motivated by the idea that although ERM-trained DNNs often take advantage of spurious features, previous works show its capability to produce good features.
CROIS consists of 2 steps method 1) train an ERM model on data without the group label information and 2) correct the classifier layer by using the features extracted from stage 1) to train a GDRO model on the data with group labels. Although in practice in the experiments the hyperparameters are fixed, for both steps the model selection can be done on the validation set, the first one optimising the aggregate accuracy and the second one the worst-group accuracy.


**Summary Of The Review:**

The idea is interesting and promising, but in the current state, I am not very convinced the work is ready to be published. The authors should revise their experiments section to address my doubts.

---

> ### Author Response · Authors · 2022-11-10
> **Author response (1/2)**
>
> Thanks for the valuable feedback. We respond to the reviewer's comments and questions below.
>
> > Fully training group labels baselines could include more recent methods such as SGDRO (non-flat version of GDRO proposed in Goel et al.).
>
> Thanks for the suggestion. We have now included a comparison with the reported result from the reference by Goel et al. mentioned by the reviewer in our revised draft (see Table 18 within Appendix E), along with several other recent baselines suggested by another reviewer. All the baseline results are taken from the respective papers. On the Waterbirds dataset, our method (CROIS) outperforms Goel et al.’s reported result (CAMEL) by 0.5% in terms of the worst-group accuracy, while using 30% of the group labels from the training split only for retraining the classifier.
>
> Notice that this result is achieved without using any data augmentation (as in CAMEL). It is conceivable that one might be able to combine our findings with data augmentation as proposed by Goel et al. to get better results. This is an interesting research question for future work. The conceptual message of our work is that one can design a simple sample splitting procedure (one portion for the feature encoder and another portion for the predictor) to learn group robust models.
>
> > There is a misleading sentence on page 3 when describing the Waterbirds dataset: “The bird images are then modified with either a water or land background.” There is no “modification” involved, in the dataset the birds are already placed either on water or land background (this is a minor note, but should be revisited).
>
> Thanks for pointing out this issue. We concur with the reviewer that the waterbird dataset is created by merging the bird images from the CUB dataset with water/land background from the PLACES dataset (as in paragraph 1 of Section 2.1 of Sagawa et al. 2020). We apologize for the confusion and have clarified this sentence in the revised version.
>
> > The main weaknesses are all in the “Experiments” section. 1) Although the hyperparameters are fixed, some ablation is still required (for example the regularisation in the second stage for the CivilComments dataset).
>
> Thanks for the comment. In Section 4.3, we have provided ablation studies by varying different fractions of $p$ while fixing the rest of the hyperparameters. We have also provided ablation studies for the learning rate and weight decay rate (see Table 12 in Appendix C). For the CivilComments dataset (unlike other datasets), the default $\ell_2$-regularization parameter is unusually high for the ERM phase; this is currently noted in Section A.2. Hence, we remove such $\ell_2$-regularization and only retrain the linear classifier layer in the second phase. We have also conducted our ablation studies (see Tables 9, 10, and 11 in Appendix C.1) that show CROIS is robust to different regularization settings. See the results below for the CivilComments dataset (same as Table 11):
>
>
> | Avg acc/Wg acc | $p=0.50$ | $p=0.30$ | $p=0.10$ |
> |---|---|---|---|
> | $\ell_2=1$ | 88.8 (1.3)/70.0 (1.6) | 89.5 (0.4)/66.4 (2.0) | 89.3 (1.7)/68.9 (2.6) |
> | $\ell_2=10^{-2}$ | 89.4 (1.0)/70.6 (0.4) | 89.5 (0.3)/68.5 (0.9) | 88.6 (1.7)/**70.2 (1.6)** |
> | $\ell_2=0$ | **89.5 (0.7)/71.0 (1.5)** | **89.7 (0.3)/68.6 (1.5)** | **89.5 (1.8)**/68.7 (1.7) |
>
>
>
>
> > 2) In Tables 1 and 2 it is not clear what the results are, is it correct to assume they are the mean and std. performances over three different initialization seeds?
>
> The reviewer is correct that in Table 1, the results correspond to the mean and the standard deviation over three different initialization seeds. We have clarified this issue in the revised paper. Additionally, we have updated the paper to include the standard deviation among three random seeds in Table 2.
>
> > 3) The results in Tables 1 and 2 are not correct, CROIS is using the validation set for training in the second stage, not only for model selection.
>
> The reviewer is correct that CROIS uses the validation split for training the classifier layer (only). Thus, our setup in Tables 1 and 2 is the same as the experimental setup of SSA. Compared with their approach, our method dispenses with pseudo-labeling while achieving competitive performance to SSA (and JTT). Our approach dramatically simplifies the second stage of prior works including JTT and CnC as we only retrain the classifier layer in the second stage.

---

> > ### Author Response · Authors · 2022-11-10
> > **Author response (2/2)**
> >
> > > For the datasets considered, the validation set has the same distribution as the test set, it does not seem correct to talk about “group shift” anymore.
> >
> > Regarding your comment about group shifts, we note that there are two kinds of shifts in the datasets considered: the shift in the mixing weights of the groups from train to validation to test, and the shift in the feature covariate distributions of different groups.
> >
> > First, notice that there is a group shift in terms of the mixing weights for both the Waterbirds and the CelebA datasets.
> >     - In the Waterbirds dataset, the number of examples in the largest group in train/validation/test are 3498/467/2255 whereas the corresponding numbers for the smallest group are 56/133/642. Notice that the ratio between the 2 groups is 62x in train but only 3.5x in test.
> >     - In the CelebA dataset, the number of samples of the largest group is train/validation/test are 71.6k/8.5k/9.7k whereas those for the smallest group are 1.4k/182/180. The ratio between the 2 groups is 47x in validation but 54x in test.
> >
> > Second, by considering the worst-group accuracy as the performance metric, the shift between different groups in the feature distribution manifests in the performance metric, since all the weight is shifted to the worst-performing group. We hope this discussion clarifies your comment.
> >
> > > 4) Table 2 should also include the average test accuracy to show the trade-off between average and worst-case performances when varying the size of the validation set. Also, why is the std. not reported here?
> >
> > We agree that understanding the tradeoff between average and worst-group accuracies would be interesting. In the revised paper, we have included the average test accuracy of our method in Table 2 (copied below). The error bars are also reported in the Table. Thanks for the suggestion.
> >
> > |  | CelebA |  |  | Waterbird |  |  |
> > |---|---|---|---|---|---|---|
> > | % of val. | 20% | 10% | 5% | 20% | 10% | 5% |
> > | JTT | 81.1 | 81.1 | 82.2 | 84.0 | 86.9 | 76.0 |
> > | SSA | 88.9 | **90.0** | 86.7 | 88.9 | **88.9** | 87.1 |
> > | CROIS (wg acc) | **89.6 (0.4)** | 87.6 (0.6) | **87.3 (1.0)** | **90.4 (1.0)** | 88.2 (0.9) | **87.8 (1.3)** |
> > | CROIS (avg acc) | 90.8 (0.2) | 91.6 (0.3) | 87.8 (1.6) | 92.4 (0.5) | 93.0 (0.7) | 88.7 (1.6) |
> >
> >
> > > 5) Table 3 shows promising results, but the best fraction “p” is a hyperparameter that should be tuned using the worst-group performance on the validation set, like the regularization term for GDRO [...]
> >
> > In our setup, we use $p$ fraction of the training split to train the feature extractor. We use the rest of the $1-p$ fraction of the training split to train the classifier/predictor.  The non-linear behavior  (observed in Table 3) comes from the fact that the feature extractor and the classifier perform very different functions. Our intuition is that the best value of p would depend on the group size ratios in the datasets; thus, we see that their values are different across different datasets. But we emphasize that this is the only parameter we need to tune in our results in Table 1 and Table 3. By comparison, prior methods all require searching many more parameters: for example, in JTT, one need to tune not only the learning rate and the weight decay rate for both phases, but also need to search across different early stopping epoch for inferring pseudo-group-labels and tune the up-sampling factor for the inferred labels.
> >
> > We would also like to point out that the sample splitting procedure is quite different from the regularization term used for GDRO. Our procedure divides the training group labels into one portion for fitting the feature extractor and another portion for fitting the classifier. We have revised the paper to clarify the sentence that the review pointed out as well.
> >
> > > 6) To obtain robust results, it would have been better to evaluate the methods across different splits of train-val-test, not simply different initialisation seeds.
> >
> > Note that all datasets/standard benchmarks in our experiments have pre-specified train-val-test splits. Therefore, it makes more sense to use the standard splits so that one can directly compare the results with the reported results from prior works in a fair manner, i.e. using the same train-val-test split (and only examining the variance across different initialization seeds).
> >
> > In Section 4.1, a similar setup to what the reviewer suggested was performed: in each run we only sample the group labels of (e.g., 10% of) the validation set. Different runs have a different 10% of the group labels in the validation split and the variance in the test accuracy of our method is small (e.g., less than 1% if we randomly sample 10% from the validation split).

---

### Author Response · Authors · 2022-11-14
**Summary of Revision**

We thank the reviewers again for your time and suggestions. We have uploaded a revision that includes the promised changes, and we hope that you could kindly take a look at the revised edition. In summary,

- We have improved our experiments by adding error bars and average accuracy to CROIS to Table 2;
- Added additional baselines (CVAR DRO, UMIX, EIIL, LFF, LISA) to the setup in Section 4.1 and 4.2 as suggested by R1 and R3. These are presented in Section E.1 and E.2 of the appendix and will be moved to the main body if more space is permitted;
- Improved the writing to clarify the use of group labels (from standard train/val/test splits) in Section 4.1, the use of parameter $p$ in Section 4.2, as well as the construction of the Waterbird dataset;
- Improved our notations in Algorithm 1 and use them throughout the paper to clarify meaning (as suggested by R2);
- Finally, we moved “models” section and Table comparing l2 regularization of GDRO vs CROIS to appendix along with some minor editing to make room for additional details/results.

We would like to thank the reviewers again for these valuable suggestions! Please let us know if you have any additional concern. We’d be happy to clarify and discuss further before the rebuttal period is due.

Update (Nov 18): We have uploaded a new revision that adds CAMEL (Goel et al.) as a baseline in the setting where group labels from the training split are available (as suggested by R1).

---

### Decision · Program_Chairs · 2023-01-20

**Decision:**

Reject

**Justification For Why Not Higher Score:**

N/A

**Justification For Why Not Lower Score:**

N/A

**Metareview: Summary, Strengths And Weaknesses:**

This paper proposes a simple yet promising method for improving worst group performance of a trained model. The idea is to use group-unlabeled training data to learn good features with an ERM objective, and then to re-train the last layer of the model on group-labeled data using a GDRO objective. The reviewers have raised multiple concerns regarding empirical evaluations and technical novelty. AC agrees with the reviewers that albeit it’s an important study, limited technical contributions and deficient empirical evidence are critical issues that will require too many revisions to be prepared and verified at this stage, and will benefit from another round of reviews. We hope the reviews are useful to improve the manuscript.